# Breast and Lung Anticancer Peptides Classification Using N-Grams and Ensemble Learning Techniques

**Ayad Rodhan Abbas *** , **Bashar Saadoon Mahdi** and **Osamah Younus Fadhil**

Department of Computer Science, University of Technology-Iraq, Baghdad 10066, Iraq; bashar.s.mahdi@uotechnology.edu.iq (B.S.M.); osamah.y.fadhil@uotechnology.edu.iq (O.Y.F.)

* Correspondence: ayad.r.abbas@uotechnology.edu.iq

**Abstract:** Anticancer peptides (ACPs) are short protein sequences; they perform functions like some hormones and enzymes inside the body. The role of any protein or peptide is related to its structure and the sequence of amino acids that make up it. There are 20 types of amino acids in humans, and each of them has a particular characteristic according to its chemical structure. Current machine and deep learning models have been used to classify ACPs problems. However, these models have neglected Amino Acid Repeats (AARs) that play an essential role in the function and structure of peptides. Therefore, in this paper, ACPs offer a promising route for novel anticancer peptides by extracting AARs based on N-Grams and k-mers using two peptides' datasets. These datasets pointed to breast and lung cancer cells assembled and curated manually from the Cancer Peptide and Protein Database (CancerPPD). Every dataset consists of a sequence of peptides and their synthesis and anticancer activity on breast and lung cancer cell lines. Five different feature selection methods were used in this paper to improve classification performance and reduce the experimental costs. After that, ACPs were classified using four classifiers, namely AdaBoost, Random Forest Tree (RFT), Multi-class Support Vector Machine (SVM), and Multi-Layer Perceptron (MLP). These classifiers were evaluated by applying five well-known evaluation metrics. Experimental results showed that the breast and lung ACPs classification process provided an accurate performance that reached 89.25% and 92.56%, respectively. In terms of AUC, it reached 95.35% and 96.92% for both breast and lung ACPs, respectively. The proposed classifiers performed competently somewhat equally in AUC, accuracy, precision, F-measures, and recall, except for Multi-class SVM-based feature selection, which showed superior performance. As a result, this paper significantly improved the predictive performance that can effectively distinguish ACPs as virtual inactive, experimental inactive, moderately active, and very active.

**Keywords:** anticancer peptides; amino acid repeats; N-Grams; machine learning; ensemble learning

## 1. Introduction

Cancer or malignant tumors are common terms for many diseases that can affect any part of the human body. Cancer has one discriminant feature: the rapid growth and production of abnormal cells that grow beyond their actual boundaries, infecting adjoining parts of the human body and spreading to other parts; this process is called metastasis. Metastases are the primary cause of millions of cancer deaths every year, especially breast and lung cancer [1]. Conventional cancer treatment procedures, such as radiotherapy and chemotherapy, are costly and frequently have harmful side effects on normal cells. Furthermore, cancer cells can develop resistance to currently available anticancer chemotherapeutic medicines [2]. As a result, developing and anticipating innovative therapies with specific mechanisms is critical. ACPs are special molecules compared to the real chemotherapy arsenal accessible to treat cancer. They show a spectrum of action patterns co-existing in some cancers [3]. Therefore, they represent an alternative to conventional chemotherapy. In addition, ACPs' activity depends on their amino acids'

type, number, and structure. They effectively destroy the cancer cell structure, thereby inhibiting the proliferation and growth of cancer cells and inducing apoptosis throughout the electrostatic interaction with the cancer cell membrane according to its structure, ACPs can be classified into four types: α-helical, β-pleated sheets, random coil, and cyclic. ACPs' design represents a challenge to researchers since they are expected to be selective to tumor cells without influencing normal body functions [4].

Traditionally, peptides can be divided into two types: oligopeptides and polypeptides, where oligopeptides have few amino acids, from 7 to 30 (e.g., A, R, N, D, C, Q, E, G, H, I, L, K, M, F, P, S, T, W, Y, and V), and includes dipeptides, tripeptides, and tetrapeptides. In contrast, polypeptides have many amino acids with more than 20 residues [5]. ACPs are divided into two categories: those toxic to both cancerous and normal cells with little indication of selectivity, and those toxic to cancer cells but not to normal mammalian cells or erythrocytes [6].

It is difficult to identify distinct ACPs quickly and effectively to understand their anticancer mechanisms better and develop new anticancer drugs. Many experimental methods for identifying and developing novel ACPs have been developed. However, they are usually time-consuming, expensive, and difficult to perform at high throughput. As a result, the enormous therapeutic importance of ACPs requires the development of highly effective prediction algorithms [7,8].

Recently, many different experimental methods have been proposed to classify APCs using either machine or deep learning.

However, there are certain limitations associated with the previous methods. The main limitation includes the type and quantity of dataset selection. In other words, previous methods focused on the binary classification of ACPs only (ACPs and non-ACPs), regardless of the ACPs type (e.g., cancer or lung). This creates a difficult situation for the experimental scientist in testing a particular type and classification of ACPs. Another important drawback of the previous methods is that AARs, which play a fundamental role in the function and structure of the peptides, are neglected, and this causes an inability to distinguish between peptides having similar structures but different activities. The issues above motivate us to propose a method that addresses the above-mentioned limitations. Therefore, this paper combines Natural Language Processing (NLP) and ensemble learning algorithms to classify breast and lung ACPs. Furthermore, we used two datasets (breast and lung cancer datasets) to perform a statistical evaluation and improve the robustness and reliability of the average performance. This paper has three contributions: first, different sizes of N-gram and k-length substrings (k-mers) were used to analyze the sequence of amino acids, specially AARs. Second, the proposed ensemble learning model improved the average prediction performance compared to that of any single machine learning model. Third, feature selection enabled ensemble learning to train faster by reducing the complexity of a model.

The rest of this paper is organized as follows. In Section 2, related works are discussed. In Section 3, the breast and lung databases are presented. In Section 3, feature extraction for peptides is illustrated. In Section 4, five feature selection methods are introduced. Section 5 provides the details of four classifiers. Section 6 demonstrates all experimental results. The last section presents the conclusions.

## 2. Related Work

This section highlights the works on ACPs classification using well-known techniques, either traditional machine or deep learning techniques. All works only classify ACPs into two labels: ACPs (positive) and non-ACPs (negative), using different datasets, as shown in Table 1.

Machine learning techniques require hand-crafted feature extraction to represent peptide sequences for classification tasks. These features are divided into five groups: binary profiles, composition-based features, structure-based features, patterns, and evolutionary

information. In contrast, the composition-based features are amino acid index, amino acid composition, dipeptide composition, and physicochemical-based features [9].

Recently, some methods in the classification of ACPs have been developed. For example, iACP describes a sequence-based predictor built by optimizing the g-gap dipeptide components. Rigorous cross-validations have proved that the new predictor has significantly outperformed existing predictors for accuracy and overall stability. Furthermore, it uses the main concept of SVM to build a separating hyper-plane that maximizes the difference between the positive and negative datasets [10].

MLACP constructed SVM and RFT algorithms for predicting ACPs from amino acid sequence parameters using the Hajisharifi dataset [11]. SAP used only 400-dimension features with g-gap dipeptide features that were collected from the Hajisharifi dataset, and then the irrelevant features were removed using the features selection method. When SVM or LibD3C are used, the 400-dimension features perform better than the RFT model [12]. mACPpred applied the feature selection approach on seven feature encodings collected from the independent dataset. These important features are then fed into an SVM classifier to develop the prediction model [13]. ACPs are used to extract a 19-dimensional feature model with lower dimensions and better performance than those of specific existing approaches. The selected features are then fed into three classifiers, SVM, RFT, and LibD3C, using the Hajisharifi dataset [14]. In cACP-2LFS, three different nature encoding approaches are used to extract essential features from peptide sequences. However, K-space amino acid pair (KSAAP) is applied to extract highly linked and valuable descriptors. In addition, a unique two-level feature selection (2LFS) technique is used to select important features and reduce the dimensionality of the proposed descriptors [15]. AntiCP 2.0 is an improved version of AntiCP, which was created to predict anticancer peptides using various input features accurately and implement the Extra Tree classifier model on two datasets: main and alternate datasets [16].

Unfortunately, these exciting techniques rely on one-experience-based, hand-crafted features, which have two limitations; The ability to represent models to a certain extent and the lack of adaptability to a different dataset, which limits the improvement of predictive performance and affects the robustness of predictive models [17]. Therefore, many works used deep learning for extracting and combining spatial features from various datasets. For example, ACP-DL constructed a deep learning long short-term memory (LSTM) neural network model to effectively predict novel ACPs using a feature representation method incorporating binary profile features and a k-mer sparse matrix [18].

**Table 1.** Summary of the related work.

| References No. | Method | Dataset | No. of Non-ACP vs. ACP | Classifier |
|---|---|---|---|---|
| [10] | iACP | Hajisharifi | 206 non-ACP 138 ACP | SVM |
| [11] | MLACP | CancerPPD | 206 non-ACP 138 ACP | SVM and RFT |
| [12] | SAP | Hajisharifi | 206 non-ACP 138 ACP | SVM |
| [13] | mACPpred | Independent dataset | 157 non-ACP 157 ACP | SVM |
| [14] | Li, Qingwen, et al. | Hajisharifi | 206 non-ACP 138 ACP | SVM, RFT, and LibD3C |
| [15] | cACP-2LFS | CancerPPD | 150 non-ACP 150 ACP | FKNN, SVM and RFT |
| [16] | AntiCP 2.0 | Main dataset | 861 non-ACP 861 ACP | Extra Tree |
| [18] | ACP-DL | ACP740 and ACP240 | 364/111 non-ACP 376/129 ACP | LSTM |
| [19] | xDeep-AcPEP | CancerPPD | 65 non-ACP 85 ACP | CNN |
| [20] | ACP-MHCNN | Independent dataset | 364/111 non-ACP 376/129 ACP | CNN |
| [21] | DLFF-ACP | CancerPPD | 65 non-ACP 85 ACP | CNN |
| [22] | ACPNet | Independent dataset | 364/111 non-ACP 376/129 ACP | RNN |

Another method, xDeep-AcPEP, identifies effective ACPs in rational peptide design for therapeutic purposes based on CNN to predict biological activity (EC50, LC50, IC50, and LD50) against six tumor cells, including those of the breast, colon, cervix, lung, skin, and prostate [19]. Finally, three methods, like ACP-MHCNN, DLFF-ACP, and ACPNet, extract and combine discriminative features from different information sources using CNN to distinguish ACPs. These methods employ three different types of peptide sequence information, peptide physicochemical properties, and auto-encoding features linking the training process [20–22].

In the context of ACPs classification, deep learning is not automatically suitable for small data situations to make accurate decisions. It might not be the best solution for not large-scale datasets.

Here, we used the biological activity data of two famous tissue types (breast and lung) from CancerPPD to train and test using the ensemble learning method to improve the classification task. In addition, we distinguished breast and lung ACPs as virtual inactive, experimental inactive, moderately active, and very active, rather than two binary classes (ACPs and non-ACPs). Consequently, this paper significantly improves the predictive performance compared with that of other state-of-the-art methods.

### 3. Datasets

This paper used two datasets of peptides pointed to breast and lung cancer. These peptides' datasets were assembled and curated manually from the Cancer Protein and Peptides Database (CancerPPD) [23] by authors in [24] to predicate biological activities such as LD50 (Lethal Dose 50%), LC50 (Lethal Concentration 50%), EC50 (Median Effective Concentration), and IC50 (Inhibitory Concentration 50%) against lung and breast cancer cells.

Experimentally, peptides' activity of anticancer was tested; only 53 peptides in the CancerPPD database possess low micromolar activity (EC50, IC50, or LC50 < 5 μM). Therefore, the authors in [24] split this anticancer activity of peptides into four classes according to μM: virtual inactive, experimental inactive, moderately active, and very active, as shown in Table 2; In other words, these classes represent the activities against breast and lung cancer cells. The breast cancer dataset contains 949 peptides, while the lung cancer dataset contains 901 peptides.

**Table 2.** Description of anticancer peptides dataset.

| Cancer Types | Class | EC50, IC50, or LC50 | No. of Peptides |
|---|---|---|---|
| breast cancer | virtual inactive | >50 μM | 750 |
| breast cancer | experimental inactive | >50 μM | 83 |
| breast cancer | moderately active | up to 50 μM | 98 |
| breast cancer | very active | ≤5 μM | 18 |
| lung cancer | virtual inactive | >50 μM | 750 |
| lung cancer | experimental inactive | >50 μM | 52 |
| lung cancer | moderately active | up to 50 μM | 75 |
| lung cancer | very active | ≤5 μM | 24 |

In the case of ties, the less active class was chosen (>50 μM). Because the CancerPPD is biased towards active peptide annotation, a set of virtual inactive peptides have been built by randomly collecting and extracting 750 alpha-helical peptides from crystalline structures recorded in the Protein Data Bank (7–30 amino acids) [24]. As a result, the total number of inactive peptides (virtual and experimental inactive) for breast and lung cancers was 833 and 803, respectively. However, this created a state of imbalance in the dataset. To overcome this problem, we used five well-known evaluation metrics: AUC, accuracy, precision, F-measures, and recall.

### 4. Features Extraction for Peptides

Amino acid repeats are abundant in every peptide sequence because these repeats reflect the inherent biological properties of the peptide [25]. In this paper, we applied an effective approach, namely k-mers, to extract these biological properties of the peptide. k-mers are primarily used with nucleotides (i.e., A, T, G, and C) in computational genomics [26], but we used k-mers with 20 amino acids. The primary approach is to break down the peptide into four different types of k-mers (k = 1, 2, 3, and 4) that represent the following amino acid profiles:

1.   A monopeptide has a single amino acid (k = 1).
2.   A dipeptide has two amino acids (k = 2).
3.   A tripeptide has three amino acids (k = 3).
4.   A tetrapeptide has four amino acids (k = 4).

The permutation index of k-mers can be represented as peptide function $P^k$. P stands for a peptide sequence of 20 amino acids. For example, if we convert a peptide sequence (FAKALAKLAKKLL) into a single character k = 1 (one amino acid), F is one of the 20 possible elements of monopeptide. Similarly, FA has 400 possible elements of dipeptides; FAK has 8000 possible elements of tripeptides. Finally, FAKA has 160,000 possible elements of tetrapeptides.

In peptides classification, N-Grams is a consecutive subsequence of the primary structure of a peptide sequence of a length n, extensively used in NLP tasks. Where n is also the number of amino acids that are extracted from each candidate peptide sequence.

Each sequence from the ACPs dataset is divided into an overlapping subsequence of 2-mers, 3-mers, and 4-mers to extract encoded N-Grams. After that, from each of these k-mers, we extract N-Grams of sizes 2, 3, and 4, as shown in Figure 1. In this paper, unigrams (n = 1) are skipped because they do not reflect any biological representations, thus, we skipped n = 1 when (k = 1).

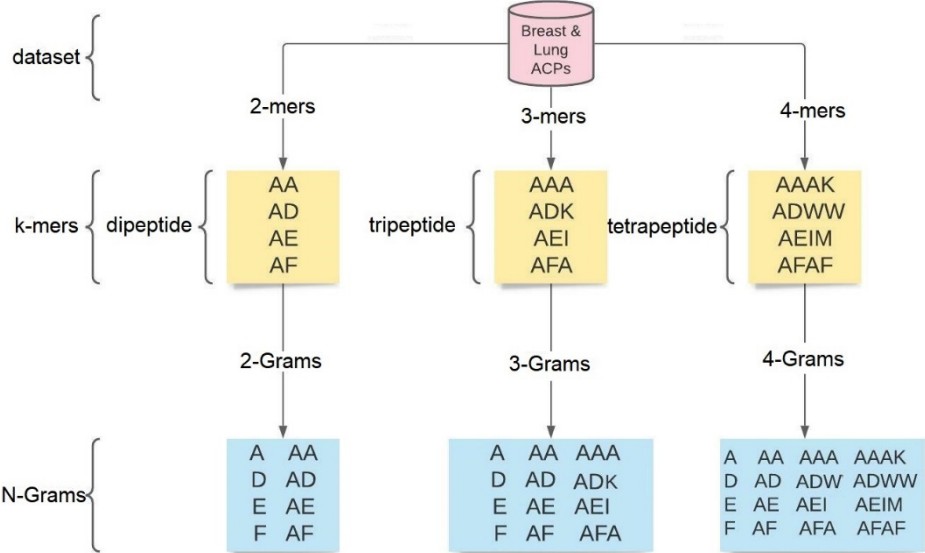

**Figure 1.** Encoded N-Grams extraction using k-mers.

The weka word2vec tool was utilized to compute vector representations of a peptide, then, the number of features extraction depended on the n values using the following formula:

$$\text{No. of features} = \sum_{k=1}^{n} p^k \tag{1}$$

For example, if we consider n = 3 (3-Grams), then the total number of peptides features of k = 1, . . . , 3, will be:

$$\text{No. of featuers} = p^1 + p^2 + p^3 \tag{2}$$

## 5. Proposed Feature Selection Methods

The major problem of the N-Grams model is that it generates features space with high dimensions and extreme spacing. In other words, most of these features have zero or empty values, and others give a number of occurrences of a given subsequence in the peptide. Furthermore, many relevant features have a high correlation degree with the class and low correlation with other features. Thus, these features provide essential information about the functions and structure of the peptide and positively affect the performance of the classifier and running time. In contrast, irrelevant or redundant features are removed from the original features vector. Therefore, in this paper, five features selection methods were successfully applied to make a comparative study of these methods and investigate the best performance of the proposed model.

These methods are Information gain [27], Gini index [28], Chi-square (X2) [29], Relief [30], and Correlation-Based Feature Selection (CBFS) [31].

## 6. Classifiers

The architecture of the proposed model consists of four stages: feature extraction, feature selection, classification, and voting process. The framework of the model is shown in Figure 2.

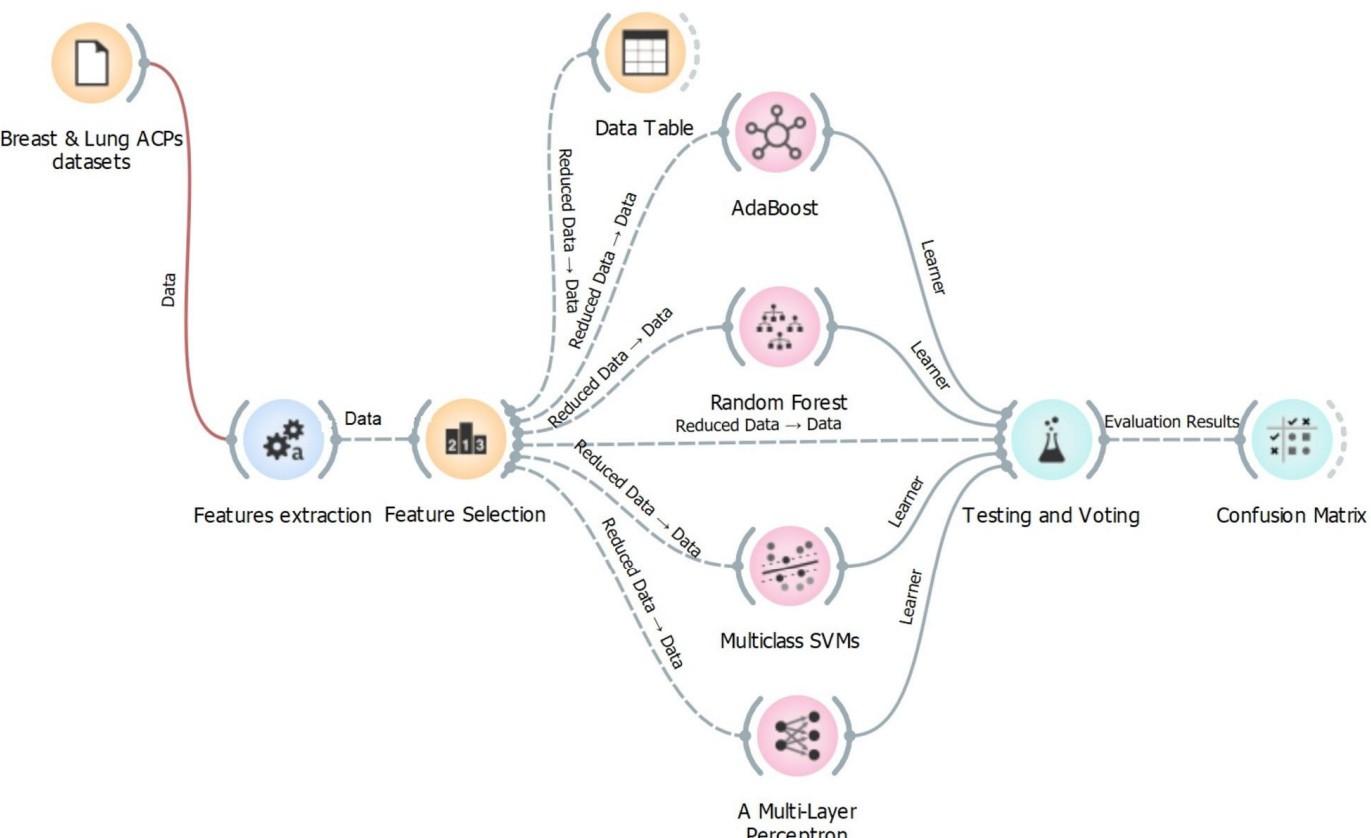

**Figure 2.** The architecture of the proposed model.

In the first stage, we extracted important features using k-mers and encoded N-Grams from each peptide sequence. Experimentally, to choose the best value for k in a k-mers model, it is necessary to find the suitable trade-off between the accuracy and the execution time. Therefore, one possible approach is to increase the value of k by one from k = 2 to

k = 4. At the same time, k = 1 or n = 1 is skipped because they do not reflect any biological representations. In contrast, we chose k = 4 since it is a common choice with large training corpora and it gives higher accuracy than other k values do, whereas a k = 2 is often used with smaller ones. Subsequently, the optimal k or n values can be considered a set leading to the highest prediction accuracy.

In the second stage, five features selection methods were successfully applied to investigate the best performance of the proposed model. Thirty-five highest ranked features out of 8420 features of the 3-Grams profile were selected using the breast ACPs dataset. Thirty-one highest ranked features out of 168,420 features of the 4-Grams profile were selected using the lung ACPs dataset. In this paper, computing time was not an issue. To get high accuracy, we considered a loop in which features were gradually (one by one) added to the prediction model depending on their importance weight (highest ranked). As a result, the best feature set could be defined as one that led to the best prediction accuracy.

In the third stage, four well-known classifiers were applied, namely, AdaBoost [32], RFT [33], Multi-class SVM [34], and MLP [35], using orange data mining software [36]. In this paper, the prediction model, 5-fold cross-validation, was used with a training size of 66% and a testing size of 33%. All experiments were run on a computer with an Intel(R) Core (TM) i5-6300U CPU 2.50 GHz using 8 GB of RAM, running Windows 10 Pro. The key parameters of each classifier are shown in Table 3.

**Table 3.** Key parameters of four classifiers.

| Classifier | Key Parameters |
|---|---|
| AdaBoost | Base estimator: Tree<br>Number of estimators: 50<br>Learning rate: 1<br>Classification algorithm: SAMME.R<br>Regression loss function: Linear |
| RFT | Number of trees:10<br>Number of attributes at each split:5<br>Limit depth of individual tree: 3<br>Don't split subset smaller than: 5 |
| Multi-class SVMs | Cost©: 1<br>Regression loss epsilon ($\varepsilon$): 0.10<br>Kernel: RBF<br>Numerical tolerance: 0.0010<br>Iteration limit: 100 |
| Multi-Layer Perceptron | Neurons in hidden layers: 100<br>Activation function: ReLu<br>Solver: Adam<br>Regularization: 0.0001<br>Maximum number of iterations: 200 |

Finally, these classifiers were combined using the voting process model to enhance the proposed model's performance by choosing the best classifier.

## 7. Results and Discussion

### 7.1. Performance Comparison with Different Amino Acid Profiles

For comparison with different N-Grams (2-Grams, 3-Grams, and 4-Grams), the average of the classification results of breast and lung ACPs using four classifiers are recorded in Figures 3 and 4, respectively. In addition, these classifiers were adopted to execute five-fold cross-validation without using feature selection methods. As we can see in Figure 3, the 3-Grams breast ACPs classification achieved the best performance, attaining an average AUC, Accuracy, F1-measures, Precision, and Recall of 94.08%, 89.28%, 88.60%, 88.19%, and 89.28, respectively. However, 2-Grams and 4-Grams had good classification results. In contrast,

the experimental metrics in Figure 4 show that the 4-Grams lung ACPs classification achieved the best performance in AUC; it was 94.94%. At the same time, 2-Grams lung ACPs classification achieved the best performance and average Accuracy, F1-measures, Precision, and Recall of 91.51%, 91.18%, 91.35%, and 91.51, respectively. However, 3-Grams had excellent classification results.

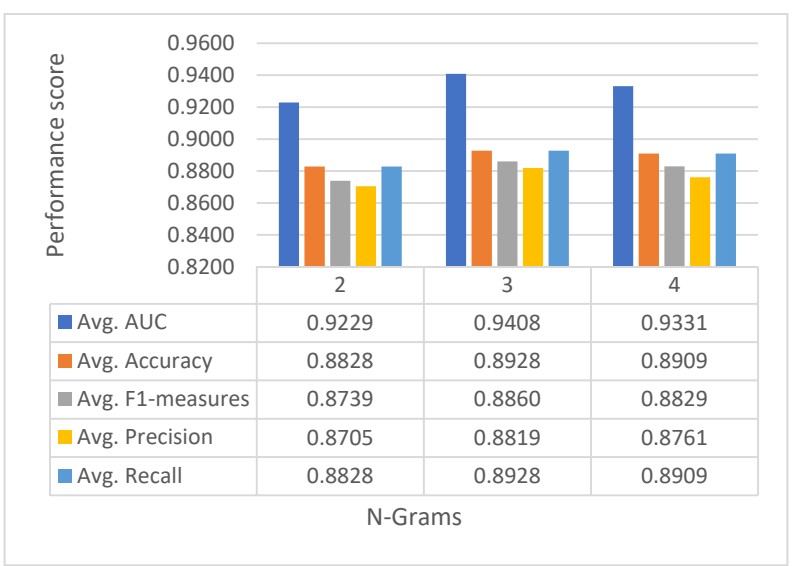

**Figure 3.** Performance evaluation of the breast ACPs classification with different N-Grams.

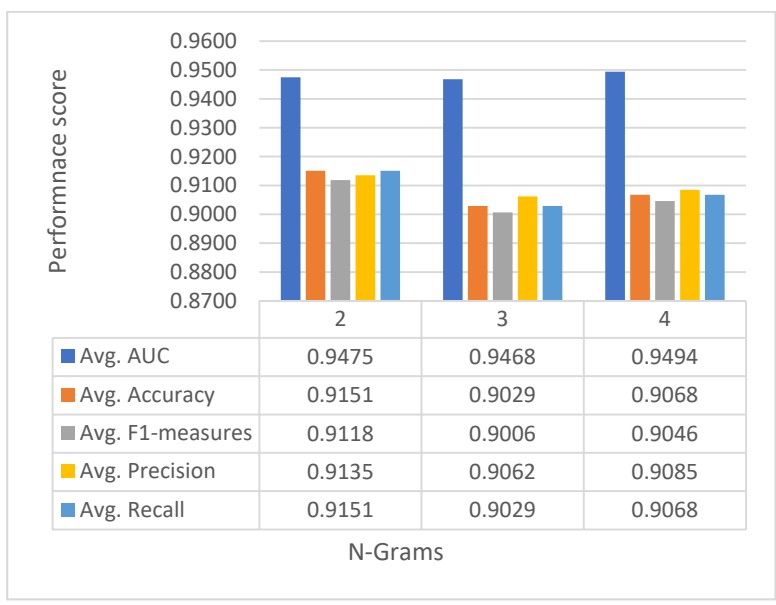

**Figure 4.** Performance evaluation of the lung ACPs classification with different N-Grams.

### 7.2. Performance Comparison with Different Features Selection Methods References

Selecting the AARs that are extracted from each candidate peptide sequence is extremely important. According to the validity of all previous experiments that directly impacted the classification process, we used the optimal values of 3-Grams and 4-Grams because they gave high AUC, Accuracy, Precision, F-measures, and Recall as they were used in large training corpora. After that, we applied five feature selection methods that used only 3-Grams and 4-Grams to facilitate comparison with these methods. Figure 5 shows the breast ACPs classification performance by selecting 35 highest ranked features out of 8420 features of the 3-Grams profile.

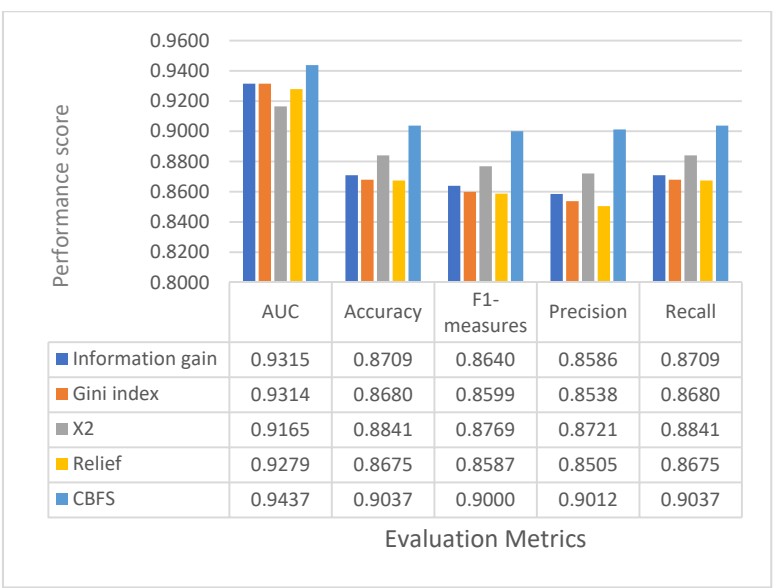

**Figure 5.** Performance of five feature selection methods on the breast ACPs using only 35 features.

Similarly, Figure 6 shows the lung anticancer peptides classification performance by selecting 31 highest ranked features out of 168,420 features of the 4-Grams profile.

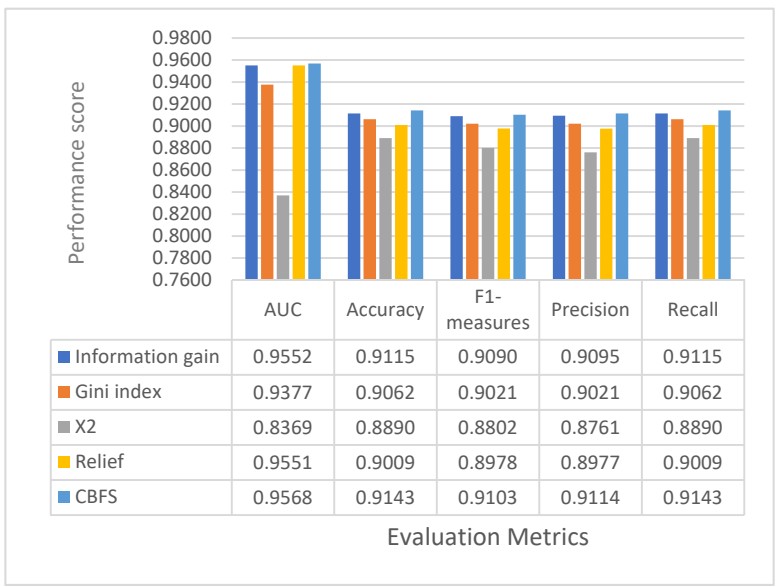

**Figure 6.** Performance of five feature selection methods on the lung ACPs using only 31 features.

All experimental results in Figure 5 show that the AUC, Accuracy, F1-measures, Precision, and Recall of the feature selection-based breast ACPs classification were about 0.29%, 1.09%, 1.4%, 1.93%, and 1.09% higher than those of the breast ACPs classification without feature selection. Similarly, all experimental results in Figure 6 show that the AUC, Accuracy, F1-measures, Precision, and Recall of the feature selection-based lung ACPs classification were about 0.74%, 0.75%, 0.57%, 0.29%, and 0.75% higher than those of the lung ACPs classification without feature selection. As a result, the average results of four classifiers on the five-fold cross-validation showed that Information gain, Gini index, X2, Relief, and CBFS were relatively stable, and CBFS had the best overall effect. Thus, optimal feature selection reached the level of best performance with 35 and 31 features using the breast and lung cancer dataset.

### 7.3. Performance Comparison with Multiple Classifiers

Because of the validity of all previous experiments that directly impacted the classification process, we chose only CBFS to classify breast and lung breast ACPs using four classifiers (AdaBoost, RFT, Multi-class SVM, and MLP). First, monopeptide, dipeptide, and tripeptide acids were extracted from the breast ACPs dataset, and we selected the 35 highest ranked features based on CBFS, as shown in Figure 7. Second, monopeptide, dipeptide, tripeptide, and tetrapeptides acids were extracted from the lung ACPs dataset, and we selected the 31 highest ranked features based on CBFS shown in Figure 8. Conventionally, these extracted features were fed to four different classifiers. In the prediction models, five-fold cross-validation was used with a training size of 66% and testing size of 33%. Multi-class SVM showed superior performance on breast ACPs dataset than other classifiers did in terms of AUC, Accuracy, F1-measures, Precision, and Recall with 95.45%, 88.62%, 87.67%, 87.96%, and 88.62, respectively. In contrast, Multi-class SVM also showed superior performance on lung ACPs compared to that of other classifier datasets in terms AUC, Accuracy, F1-measures, Precision, and Recall with 96.92%, 92.56%, 92.12%, 92.60%, and 92.65, respectively. Overall, the above analysis indicates that all four classifiers' performances were similar regardless of the datasets.

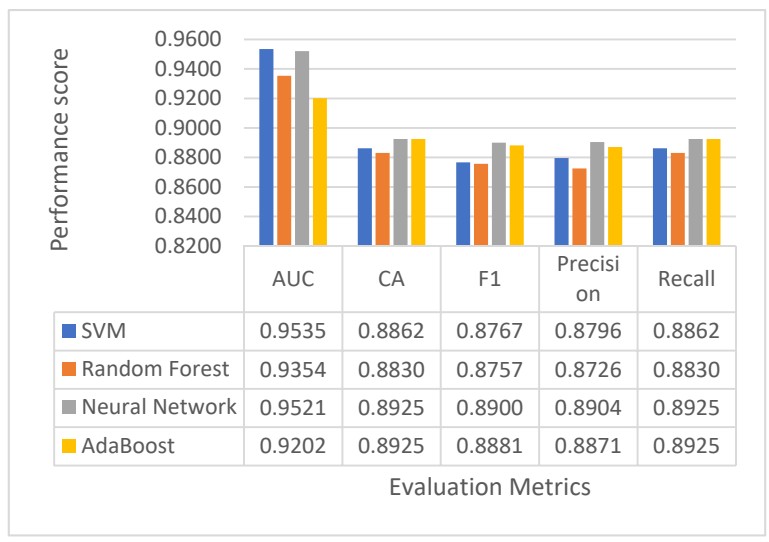

| | AUC | CA | F1 | Precision | Recall |
|---|---|---|---|---|---|
| SVM | 0.9535 | 0.8862 | 0.8767 | 0.8796 | 0.8862 |
| Random Forest | 0.9354 | 0.8830 | 0.8757 | 0.8726 | 0.8830 |
| Neural Network | 0.9521 | 0.8925 | 0.8900 | 0.8904 | 0.8925 |
| AdaBoost | 0.9202 | 0.8925 | 0.8881 | 0.8871 | 0.8925 |

**Figure 7.** Performance of four classifiers using the breast ACPs.

### 7.4. Performance Comparison with State of the Art

This section compares the proposed method with MLACP, cACP-2LFS, xDeep-AcPEP, and DLFF-ACP as state of the art [11,15,19,21], as shown in Table 4. It was not easy to directly compare the previous methods for two reasons. First, all mentioned methods in Section 2 did not consider any cancer types (e.g., breast or lung cancer). The other reason was that the purposes of these methods were almost identical, focusing only on the classification of ACPs into binary classes (non-ACP and ACP). However, we compared previous works, which used the same database (CancerPPD). In terms of accuracy, the experimental results showed as follows:

1.  The proposed breast ACPs method outperformed all the mentioned methods, except for cACP-2LFS, because a minimal dataset was used (150 non-ACP 150 ACP) with 20 features only.
2.  The proposed lung ACPs method outperformed all mentioned methods.

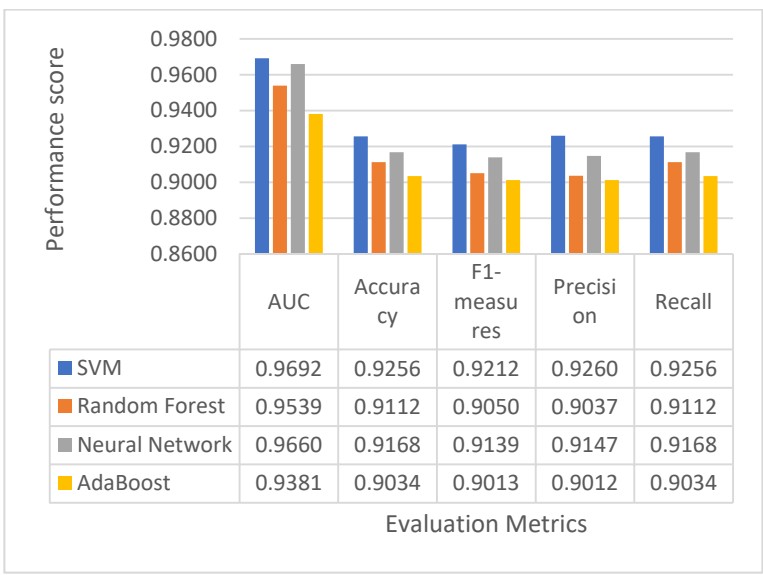

**Figure 8.** Performance of four classifiers using the lung anticancer peptides.

**Table 4.** Performance comparison with the state of the art.

| Method | Classifier | No. of Features | Accuracy | Predictions |
|---|---|---|---|---|
| MLACP | SVM and RFT | 20, 400, 5, and 11 | 88.7% | 206 non-ACP 138 ACP |
| cACP-2LFS | FKNN, SVM and RFT | 20 | 93.72% | 150 non-ACP 150 ACP |
| xDeep-AcPEP | CNN | - | 82.42% | 65 non-ACP 85 ACP |
| DLFF-ACP | CNN | - | 82% | 65 non-ACP 85 ACP |
| Proposed breast ACPs | Ensemble learning | 35 | 89.25% | 750 virtual inactive, 83 experimental inactive, 98 moderately active, and 18 very active |
| Proposed lung ACPs | Ensemble learning | 31 | 95.35% | 750 virtual inactive, 52 experimental inactive, 75 moderately active, and 24 very active |

In this paper, we used the biological activity data on two famous tissue types (breast and lung) and distinguished breast and lung ACPs as virtual inactive, experimental inactive, moderately active, and very active, rather than two binary classes (ACPs and non-ACPs).

As a result, the classification outcomes reveal that our proposed method achieved improved performance accuracy compared to that of the state of the art so far.

## 8. Conclusions

This paper proposes a new model that combines NLP and ensemble learning algorithms to efficiently and accurately classify breast and lung ACPs. Specifically, we offer a novel feature extraction method by extracting AARs using N-Grams. Then, five feature selection methods were used to improve classification performance and reduce the experimental costs. Finally, we utilized four classifiers, AdaBoost, RFT, Multi-class SVM, and MLP, and applied them on two datasets. Experimental results reveal that (1) The 3-Grams profile (monopeptide, dipeptide, and tripeptide acids) of the breast ACPs classification achieved the best performance relative to that of other profiles. (2) The 4-Grams profile (monopeptide, dipeptide, tripeptide, and tetrapeptides acids) of the lung ACPs classification achieved the best performance compared to that of other profiles in terms of AUC. At the same time, 2-Grams also achieved the best performance but with other classification matrices. (3) All feature selection methods like Information gain, Gini index, $X^2$, Relief, and CBFS were relatively stable, and CBFS had the best overall effect in breast and lung

cancer. (4) The performances of the four classifiers were similar regardless of the datasets, but Multi-class SVM-based CBFS showed superior performance in the breast and lung classification. As a result, the impact on classification performance depended on four factors: the size of N-Grams, feature selection methods, the number of selected features, and machine learning algorithms. Thus, this paper significantly improves the predictive performance that can effectively distinguish ACPs as virtual inactive, experimental inactive, moderately active, and very active.

**Author Contributions:** Conceptualization, A.R.A.; methodology, A.R.A.; software, A.R.A.; validation, A.R.A., B.S.M. and O.Y.F.; formal analysis, A.R.A. and B.S.M.; investigation, A.R.A.; resources, A.R.A.; data curation, A.R.A.; writing original draft preparation, A.R.A.; writing review and editing, A.R.A., B.S.M. and O.Y.F.; visualization, A.R.A., B.S.M. and O.Y.F.; supervision, A.R.A.; project administration, A.R.A.; funding acquisition, A.R.A., B.S.M. and O.Y.F. All authors have read and agreed to the published version of the manuscript.

**Funding:** This research received no external funding.

**Institutional Review Board Statement:** Not applicable.

**Informed Consent Statement:** Not applicable.

**Data Availability Statement:** Data supporting reported results can be found at: http://crdd.osdd.net/raghava/cancerppd/ (accessed on 1 January 2021).

**Conflicts of Interest:** The authors declare no conflict of interests.

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
