# Peer review of "Breast and Lung Anticancer Peptides Classification Using N-Grams and Ensemble Learning Techniques"

_2504-2289, doi:10.3390/bdcc6020040_

Round 1

Reviewer 1 Report

In this manuscript, the author tried and tested several feature selection and machine learning methods to classify anticancer peptides of breast and lung cancer. I have some questions and concerns:

  1. More description of the function of ACPs is required in Introduction.
  2. No need to introduce the feature selection and machine learning methods as in Sec. 4 and 5 unless the attributes of an method facilitate the application of this algorithm on the dataset used.
  3. In Line 243, how the top 35 features were selected? I guess different features should have be selected by different method. Besides, at what criterion they are selected? So does the 31 features for lung cancer ACPs.
  4. The results were poorly presented and discussed, especially of the comparison of various classifiers.
  5. What's the split ratio of training and test dataset?
  6. It looks the sample size is less than 2000 while there are 100 hidden layers in the MLP model? How many neurons in each layer?
  7. In Line 295, the author declares that their work "significantly improves the predictive performance", but compared with any other work?

Author Response

Dear Reviewer

Thank you for giving me the opportunity to submit a revised draft of my manuscript titled “Breast and Lung Anticancer Peptides Classification using N-Grams and Ensemble Learning Techniques” to “Big Data and Cognitive Computing”. We appreciate the time and effort that you and the reviewers have dedicated to providing your valuable feedback on my manuscript. We are grateful to the reviewers for their insightful comments on my paper. We have been able to incorporate changes to reflect most of the suggestions provided by the reviewers. We have highlighted the changes within the manuscript.

Comments from Reviewer 1:

Comment 1: More description of the function of ‎ACPs is required in Introduction.‎

Response:  Thank you for pointing this out. We agree with this comment. Therefore, we have done and added some information and function about ACPs, this change can be found in page 1 line 39-43, page 2 line 45-52, page 2 line 56-67, and page 3 line 98-107.

Comment 2: ‎ No need to introduce the feature selection and machine learning methods as in Sec. ‎‎4 and 5 unless the attributes of a method facilitate the application of this algorithm ‎on the dataset used.‎

Response:  Thank you for pointing this out. We agree with this comment. Therefore, we have made some changes can be found in pages 7 and 10.

Comment 3: In Line 243, how the top 35 features were selected? I guess different features should have be selected by a different method. Besides, at what criterion are they selected? So does the 31 features for lung cancer ACPs.

Response:  Thank you for pointing this out. In this paper, to get high accuracy, we consider a loop ‎in which features are gradually (one by one) added to the prediction model depending ‎on their importance weight (highest ranked). As a result, the best feature set can be defined as one that leads to the best prediction accuracy.‎ This answer can be found in page 8 line 306-309.

Comment 4: The results were poorly presented and discussed, especially of the comparison of various classifiers

Response:  Thank you for pointing this out. Some changes have been made in page 11 line 385-390, page 12 line 408-410, and page 13 line 422-427.

 Comment 5: What's the split ratio of training and test dataset?

Response:  The answer can be found in page 8 line 313 and page 13 line 422.

Comment 6: It looks the sample size is less than 2000 while there are 100 hidden layers in the MLP model? How many neurons in each layer?

Response: Thank you for pointing this out. In page 9, table 3 show the key parameters of Multi-Layer Perceptron. The number of neurons in hidden layers= 100.

Comment 7: In Line 295, the author declares that their work "significantly improves the predictive performance", but compared with any other work?

Response: Thank you for pointing this out. We agree with this comment. Therefore, we add the performance comparison with state-of-the-art. This can be found in page 14 line 433-454.

Reviewer 2 Report

The authors propose classifying anti-cancer peptides using a novel ensemble method. They also claim that this method accounts for amino acid repeats, which have been neglected in other models. First of all, the relevance of this classification has not been clearly presented by the authors. What do "virtual inactive", "experimental inactive", "moderately active", and "very active" represent, and how would this model/method be useful in cancer interventions? Does this also predict if said peptide is an ACP? Secondly, I am concerned about how imbalanced the training set is. Nearly 80% of the training data belongs to one class. Thirdly, while the authors claim to present an ensemble classification method, I am just seeing comparison of 4 different classifiers in the results section. They do not seem be combined together. Fourthly, I would like to have seen a comparison of the results with already developed classification methods. The advantages of this method are not adequately explained.

I am also sharing some of the more specific comments that I noted down while reviewing this manuscript:

Page 1 14-15: "anticancer peptides" instead of "anticancer", "two peptide datasets" instead of "two peptides' datasets"
Page 1 38-40: "Whereas ACPs are special molecules compared to the real chemotherapy arsenal accessible to treat cancer. ACPs show a spectrum of action patterns co-existing in some cancers [2]." - suggest rewording, do you mean comparable instead of compared?
Page 2 53: When you say "predict ACPs", do they just predict whether they are ACPs or not? Or do they also classify?
Page 2 63-64: What is AA-index? Reference needed.
Page 2 66-67: "Unfortunately, these exciting techniques suffer from experience-based, hand-crafted features, which have two limitations" - suggest reword "suffer from" to "rely on"
Page 2 81-82: Does your ensemble method perform better than the existing ensemble methods?
Page 2 85-86: "databases" or "datasets"?
Page 2 94-95: A little explanation of what these classes mean would be helpful
Page 2-3 95-98: Your training set does not seem very balanced. Nearly 80% is virtual inactive.How are you handling this? Are you doing undersampling for the majority class?
Page 3 100-101: "Amino acid repeats are abundant in every peptide sequence because these repeats reflect the inherent biological properties of the peptide." - references?
Page 3 104-106: Why did you pick a max k-mer size of 4?
Page 3 117-119: "In peptides classification, N-Grams is a consecutive subsequence of the primary structure of a peptide sequence of a length n, extensively used in NLP tasks. In contrast, n is also the number of amino acids that are extracted from each candidate peptide sequence" - how is this "in contrast"?
Page 3 Figure 1: How do you get AAW in your 4-grams? N-grams are supposed to be consecutive subsequences? 
Page 5 177: "orange data mining software" - reference?
Page 6 203-212: wrong description for multiclass SVM

Thanks and best of luck.

Author Response

Dear Reviewer

Thank you for giving me the opportunity to submit a revised draft of my manuscript titled “Breast and Lung Anticancer Peptides Classification using N-Grams and Ensemble Learning Techniques” to “Big Data and Cognitive Computing”. We appreciate the time and effort that you and the reviewers have dedicated to providing your valuable feedback on my manuscript. We are grateful to the reviewers for their insightful comments on my paper. We have been able to incorporate changes to reflect most of the suggestions provided by the reviewers. We have highlighted the changes within the manuscript.

Comments from Reviewer 2:

Comment 1: Page 1 14-15: "anticancer peptides" instead of "anticancer", "two peptide datasets" instead of "two peptides' datasets".

Response:  Thank you for pointing this out. We agree with this comment. Therefore, this change has been done. It can be found in page 1 line 14-15.

Comment 2: ‎ Page 1 38-40: "Whereas ACPs are special molecules compared to the real chemotherapy arsenal accessible to treat cancer. ACPs show a spectrum of action patterns co-existing in some cancers [2]." - suggest rewording, do you mean comparable instead of compared?

Response:  You have raised an important point here. However, we believe that “compared “would be more appropriate Ref. [3].

Comment 3: Page 2 53: When you say "predict ACPs", do they just predict whether they are ACPs or not? Or do they also classify?

Response:  Thank you for this question, all previous works only classify ACPs into two labels: ACPs (positive) and non-ACPs (negative) ‎, using different datasets.‎ This answer can be found in page 3 line 98-101.

Comment 4: Page 2 63-64: What is AA-index? Reference needed.

Response: Thank you for this suggestion. In fact, the paragraph contains AA-index has been removed according to reviewers’ comments.

Comment 5: Page 2 66-67: "Unfortunately, these exciting techniques suffer from experience-based, hand-crafted features, which have two limitations"  suggest reword "suffer from" to "rely on"

Response:  Thank you for pointing this out. We agree with this and have done. It can be found in page 4 line 156.

Comment 6: Page 2 81-82: Does your ensemble method perform better than the existing ensemble methods?

Response: Yes, it does. It can be found in Page 14 line 433-454.

Comment 7: Page 2 85-86: "databases" or "datasets"?

Response: It is a database, but peptides datasets were assembled and curated manually from ‎Cancer Protein and Peptides Database (CancerPPD)‎.

Comment 8: Page 2 94-95: A little explanation of what these classes mean would be helpful.

Response: Thank you for pointing this out. These classes represent the activities against breast ‎and ‎lung cancer cells.‎ The change has been made according to this question; it can be found in ‎page 5 line 192-196.‎

Comment 9: Page 2-3 95-98: Your training set does not seem very balanced. Nearly 80% is virtual inactive. How are you handling this? Are you doing under sampling for the majority class?

Response: Thank you for pointing this out.‎ Page 5 line 197-204 Because the CancerPPD is biased towards active peptide annotation, a set of virtual ‎inactive peptides have been built by randomly collecting and extracting 750 alpha-‎helical peptides from crystalline structures recorded in the Protein Data Bank.‎ To overcome this problem, we used five well-known evaluation metrics: AUC, ‎accuracy, precision, F-measures, ‎and recall.‎ 

Comment 10: Page 3 100-101: "Amino acid repeats are abundant in every peptide sequence because these repeats reflect the inherent biological properties of the peptide." - references?

Response: We have inserted the reference [25].‎ Page 6 line 213.

Comment 11: Page 3 104-106: Why did you pick a max k-mer size of 4?

Response:  You have raised an important point here, In Page 8 line 294-301, we choose ‎k=4 since it is a common choice with large training corpora and it gives high ‎accuracy than other k, whereas a k=2 is often used ‎with smaller ones.

Comment 12: Page 3 117-119: "In peptides classification, N-Grams is a consecutive subsequence of the primary structure of a peptide sequence of a length n, extensively used in NLP tasks. In contrast, n is also the number of amino acids that are extracted from each candidate peptide sequence" - how is this "in contrast"?

Response:  Thank you for pointing this out. the change has been made in page 6 line 230-231.

Comment 13: Page 3 Figure 1: How do you get AAW in your 4-grams? N-grams are supposed to be consecutive subsequences?

Response:  Thank you for pointing this out.  for example, if the peptide sequence “AAWKWAWAKKWAKAKKWAKAA” then the 4- gram is ‎AAWK, 3-gram AWW, 2-gram AA. There was a typo mistake. The change has been made in page 6.

Comment 14: Page 5 177: "orange data mining software" - reference?

Response:  We have inserted the reference [36].‎ Page 8 line 311-312.

Comment 15: : Page 6 203-212: wrong description for multiclass SVM‎.

Response:  Thank you for this suggestion. In fact, the introduction about SVM has been removed according to reviewers’ comments.

Reviewer 3 Report

In this paper, authors study the problem of classifying Anticancer peptides ACPs ‎problems. Specifically, ACPs offer a promising route for novel anticancer ‎by extracting AARs ‎based on N-Grams ‎and k-mers using two peptides’ ‎datasets. These datasets pointed to breast and ‎lung ‎cancer cells ‎assembled and curated manually from CancerPPD‎. Every dataset consists of ‎a ‎sequence of peptides and their synthesis and anticancer activity on breast and lung cancer cell lines. ‎Five different feature selection methods ‎are used in this paper to improve classification performance ‎and reduce the ‎experimental costs. After that, ACPs are classified using four classifiers, ‎namely ‎AdaBoost, Random Forest Tree (RFT), Multi-class Support Vector ‎Machine (SVM), and Multi-Layer ‎Perceptron (MLP).

In order to better understand the outcome of this work, authors evaluate these classifiers by applying five well-known evaluation metrics. ‎Experimental results showed that the breast and lung ACPs classification process provides an ‎accurate performance that reaches ‎‎89.25% and 92.56%, respectively. In terms of AUC, it reaches ‎‎95.35% and 96.92% for both breast ‎and lung ACPs, respectively.‎ As a result, authors state that their proposed paper ‎significantly improves the predictive performance that can effectively distinguish ACPs as virtual ‎inactive, experimental inactive, moderately active, and very active.‎

The paper seems thorough with numerous details. Nevertheless, there are many points that need to be clarified before the paper is better improved.

More specifically, I think that the Introduction section should ideally be enriched so as to further explain to users who are not very familiar with this field, the purpose of appreciating the precise contribution made by this paper. What is the actual purpose of the paper that authors try to prove? Moreover, there seems to be missing a whole section, namely related work. Please elaborate.

Section 2 contains information regarding the datasets. Authors should add some tables with more specific details regarding them. There cannot be a section with only one paragraph.

Sections 3 and 4 discuss well-known information regarding data mining and information retrieval aspects. There is no comparison with other methods. How are authors differentiating from other works?

In addition, the implementation details (PC configuration, etc) must be placed.

Finally, authors should add a paragraph regarding their future work.

Author Response

Dear Reviewer

Thank you for giving me the opportunity to submit a revised draft of my manuscript titled “Breast and Lung Anticancer Peptides Classification using N-Grams and Ensemble Learning Techniques” to “Big Data and Cognitive Computing”. We appreciate the time and effort that you and the reviewers have dedicated to providing your valuable feedback on my manuscript. We are grateful to the reviewers for their insightful comments on my paper. We have been able to incorporate changes to reflect most of the suggestions provided by the reviewers. In addition, we have highlighted the changes within the manuscript.

Comments from Reviewer 3:

Comment 1: More specifically, I think that the Introduction section should ideally be enriched so as to further explain to users who are not very familiar with this field.

Response:  Thank you for pointing this out. We agree with this comment. Therefore, we have done and added some information and function about ACPs, this change can be found in page 1, line 39-43, page 2 line 45-52, page 2, line 56-67, and page 3, line 98-107.

Comment 2: ‎ The purpose of appreciating the precise contribution made by this paper. What is the actual purpose of the paper that authors try to prove?

Response:  You have raised an important point here. ‎We have changed it. This can be found in page 3, line 109-114, and page 5, line 175-180.

Comment 3: There seems to be missing a whole section, namely related work. Please elaborate.

Response:  Thank you for pointing this out. We agree with this and have done. The related work can be ‎found in page 3.‎

Comment 4: Section 2 contains information regarding the datasets. Authors should add some tables with more specific details regarding them. There cannot be a section with only one paragraph.

Response:  Thank you for pointing this out. We agree with this. We have done. It can be ‎found in page 5, line 181-207.‎

Comment 5: Sections 3 and 4 discuss well-known information regarding data mining and information retrieval aspects. There is no comparison with other methods. How are authors differentiating from other works?

Response:  Thank you for pointing this out. We agree with this comment. Therefore, we add the performance comparison with state-of-the-art. This can be found in page 14 line 433-454.

Comment 6: In addition, the implementation details (PC configuration, etc) must be placed.

Response:  Thank you for pointing this out. We agree with this comment. Therefore, we have added the PC configuration. This can be found in page 8 line 307-311.

Comment 7: Finally, authors should add a paragraph regarding their future work.

Response:  Thank you for pointing this out. We agree with this comment. Therefore, we add future work. This can be found in page 15 line 467-469.

Round 2

Reviewer 1 Report

I have no further questions.

Reviewer 3 Report

Authors have addressed my comments so I vote for acceptance!